# The role of ventromedial prefrontal cortex in reward valuation and future thinking during intertemporal choice

Elisa Ciaramelli[1,2]*, Flavia De Luca[1,2†], Donna Kwan[3], Jenkin Mok[3], Francesca Bianconi[2], Violetta Knyagnytska[2,3], Carl Craver[4], Leonard Green[5], Joel Myerson[5], R Shayna Rosenbaum[3,6]*

[1]Dipartimento di Psicologia, Università di Bologna, Bologna, Italy; [2]Centro studi e ricerche in Neuroscienze Cognitive, Università di Bologna, Cesena, Italy; [3]Department of Psychology, York University, Toronto, Canada; [4]Department of Philosophy, Washington University, St. Louis, United States; [5]Department of Psychological and Brain Sciences, Washington University, St. Louis, United States; [6]Rotman Research Institute, Baycrest, Toronto, Canada

**Abstract** Intertemporal choices require trade-offs between short-term and long-term outcomes. Ventromedial prefrontal cortex (vmPFC) damage causes steep discounting of future rewards (delay discounting [DD]) and impoverished episodic future thinking (EFT). The role of vmPFC in reward valuation, EFT, and their interaction during intertemporal choice is still unclear. Here, 12 patients with lesions to vmPFC and 41 healthy controls chose between smaller-immediate and larger-delayed hypothetical monetary rewards while we manipulated reward magnitude and the availability of EFT cues. In the EFT condition, participants imagined personal events to occur at the delays associated with the larger-delayed rewards. We found that DD was steeper in vmPFC patients compared to controls, and not modulated by reward magnitude. However, EFT cues downregulated DD in vmPFC patients as well as controls. These findings indicate that vmPFC integrity is critical for the valuation of (future) rewards, but not to instill EFT in intertemporal choice.

*For correspondence:
elisa.ciaramelli@unibo.it (EC);
shaynar@yorku.ca (RSR)

Present address: †School of Psychology, University of Sussex, Falmer, United Kingdom

Competing interests: The authors declare that no competing interests exist.

## Introduction

Choices are often intertemporal, requiring trade-offs between short-term and long-term outcomes. Human and non-human animals tend to prefer smaller-immediate over larger-delayed rewards (*Green and Myerson, 2004*; *Rudebeck et al., 2006*). This phenomenon reflects delay discounting (DD), the decrease in subjective value of a reward as the delay until its receipt increases. Several clinical conditions, such as drug addiction and obesity, are associated with steep DD (*Bickel et al., 2014*), a disproportionate prioritization of immediate gratification associated with poor self-control and impulsivity. The neural mechanisms governing DD and its adaptive modulation are thus of theoretical and clinical relevance.

The ventromedial prefrontal cortex (vmPFC) is causally implicated in intertemporal choice. Indeed, patients with damage to the vmPFC (*Sellitto et al., 2010*; *Peters and D'Esposito, 2016*; *Lq et al., 2020*; but see *Fellows and Farah, 2005*), and animals with lesions in homologous regions (*Rudebeck et al., 2006*), show abnormally steep DD. The specific role played by vmPFC in DD, however, is still debated. According to a prominent model of intertemporal choice (*Hare et al., 2009*; *Figner et al., 2010*; *Peters and Büchel, 2011*), vmPFC is engaged in reward valuation and integrates different outcome attributes (e.g., amounts, delays), whereas lateral prefrontal cortex modulates vmPFC subjective value signals to promote self-control and future-oriented choice.

In separate work, *Peters and Büchel, 2010* (see also *Benoit et al., 2011*) have shown that cues to imagine personal future events (episodic future thinking [EFT]; *Suddendorf and Corballis, 1997*; *Atance and O'Neill, 2001*) during intertemporal choices reduce DD, and the DD reduction relates to functional coupling of medial prefrontal regions and the hippocampus, and to the vividness of imagined events. This finding points to prospection as another component process of DD (*Peters and Büchel, 2011*). Indeed, EFT effects on DD were not (*Palombo et al., 2015*) or were inconsistently (*Kwan et al., 2015*) detected in amnesic patients with medial temporal lobe (MTL) lesions, in line with the evidence that MTL patients cannot imagine detail-rich future events (*Race et al., 2011*).

In addition to its role in reward valuation (*Bartra et al., 2013*), the vmPFC is also a crucial substrate of prospection (*Schacter et al., 2012*), and, accordingly, vmPFC patients are impaired in EFT (*Bertossi et al., 2016a*; *Bertossi et al., 2016b*; *Bertossi et al., 2017*; *Verfaellie et al., 2019*). Although MTL patients are as well, recent research suggests that the nature of the EFT impairment is different in each case. Whereas constructed experience in hippocampal patients is mainly devoid of spatial references, that of vmPFC patients also lacks relevant contents and sensory details, suggesting that vmPFC plays a more general (upstream) role in event construction (*De Luca et al., 2018*). *McCormick et al., 2018*, therefore, have proposed that vmPFC initiates (future) event construction by activating schematic knowledge (e.g., about the self, lifetime periods) that drives the collection of relevant individual details, which the hippocampus then assembles into spatially coherent scenes (see also *Ciaramelli et al., 2019*; *D'Argembeau, 2020*; *Moscovitch et al., 2016*; *Ghosh et al., 2014*). DD, therefore, could be causally linked to vmPFC through prospection, as well as through its role in reward valuation.

To investigate the specific contributions of vmPFC to both the reward valuation and prospection components of DD, we compared the effects of reward magnitude and EFT on DD on patients with vmPFC damage with their effects on healthy controls. Twelve vmPFC patients (see *Figure 1* for the extent and overlap of vmPFC patients' lesions) and 41 healthy controls matched to patients in age

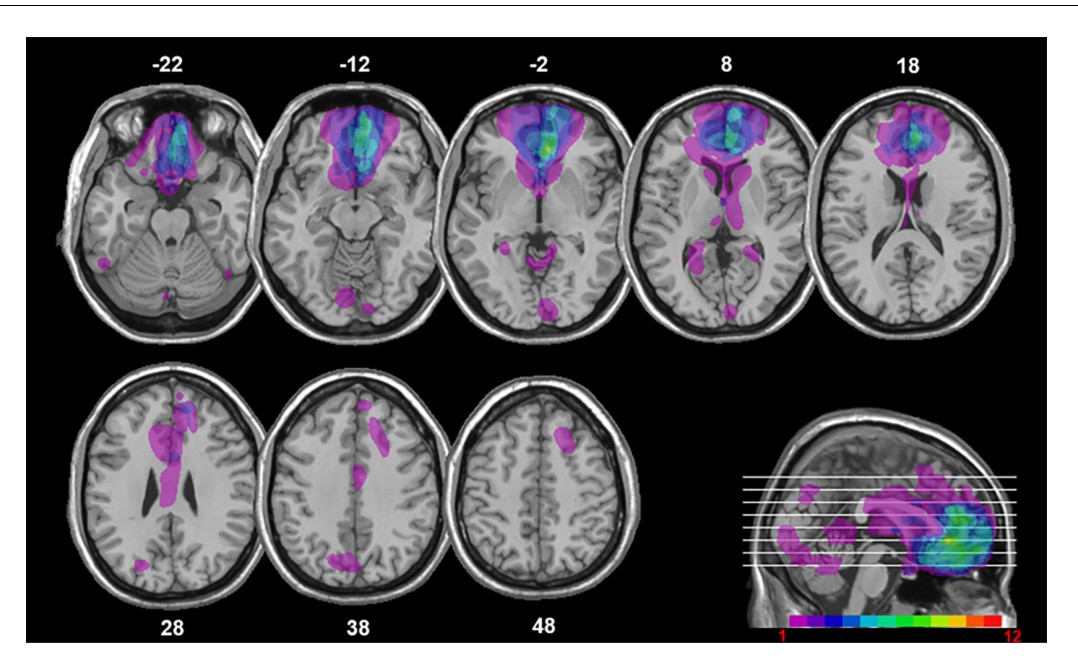

**Figure 1.** Location and overlap of brain lesions. The panel shows the lesions of the 12 patients with ventromedial prefrontal cortex (vmPFC) damage projected on the same eight axial slices and on the mesial view of the standard Montreal Neurological Institute (MNI) brain. The level of the axial slices is indicated by horizontal lines on the mesial view of the brain, and by z-coordinates. The color bar indicates the number of overlapping lesions. Maximal overlap occurs in Brodmann areas (BAs) 11, 10, and 32 of vmPFC. In axial slices, the left hemisphere is on the left side.

(57.41 vs. 61.09; $t_{51}$ = 1.61; p = 0.11), education (13.41 vs. 13.19; $t_{51}$ = −0.22; p = 0.82), and gender balance (8 males, 35 males; $\chi^2$ = 2.12; p = 0.14) chose between smaller-immediate and larger-delayed rewards while we manipulated reward magnitude (small magnitude: €80/$100; large magnitude: €1500/$2000) and the availability of EFT cues during intertemporal choices. In the EFT condition, participants were presented with subject-specific cues to imagine personal future events to occur at the delays associated with the larger-delayed rewards (see *Figure 2* for an example trial).

A magnitude effect is consistently observed such that people discount larger rewards less steeply than smaller rewards (*Green et al., 1997*). This effect has been ascribed to self-control mechanisms supported by the lateral prefrontal cortex (e.g., *Ballard et al., 2017*). However, impaired reward valuation following vmPFC damage should hinder the differential valuation of large vs. small rewards, and prevent the implementation of self-control when large rewards are at stake. Thus, we predict, in addition to steep DD, a smaller magnitude effect in vmPFC patients compared to healthy controls.

Concerning prospection, previous studies have observed an EFT effect on DD, such that people discount future rewards less steeply if cued to imagine personal future events during intertemporal choice (*Peters and Büchel, 2010*; *Benoit et al., 2011*). Considering that vmPFC is implicated in prospection (*Schacter et al., 2012*), and that vmPFC patients are impaired in EFT (*Bertossi et al., 2016a*; *Bertossi et al., 2016b*; *Bertossi et al., 2017*), vmPFC patients' DD should remain steep even when EFT cues are provided, because patients may nevertheless fail to construct the vivid future events that might be needed to counteract DD. Thus, we predict a reduced EFT effect on DD in vmPFC patients compared to healthy controls.

## Results

### DD rates

Preliminary fits of the hyperbolic function SV = 1/(1+kD), with SV = subjective value (expressed as a fraction of the delayed amount) and D = delay (in days), to individual participants' data using a non-linear least-squares algorithm (implemented in Statistica Statsoft) revealed that subjective preferences were not equally well characterized by hyperbolic functions in the Standard and EFT conditions. This was especially apparent in vmPFC patients in the EFT condition whose discounting curves were not always monotonically decreasing (see *Figure 3*, and *Figure 3—figure supplement 1* for individual patients' discounting curves).

The degree to which participants discounted delayed rewards (DD rate), therefore, was measured using the area under the curve (AuC), a theoretically neutral, normalized measure of DD that does not depend on theoretical models regarding the shape of the discounting curve (*Myerson et al., 2001*). *Figure 4* displays the AuC by participant group and condition, as well as individual participants' data. An ANOVA on AuCs with Group (vmPFC patients, healthy controls), Condition (Standard, EFT), and Reward magnitude (small, large) as factors revealed an effect of Reward magnitude ($F_{1,51}$ = 13.17, p = 0.0007, partial $\eta^2$ = 0.20), qualified by a Group × Reward magnitude interaction ($F_{1,51}$ = 9.49, p = 0.003, partial $\eta^2$ = 0.16). Fisher post hoc tests showed that healthy controls

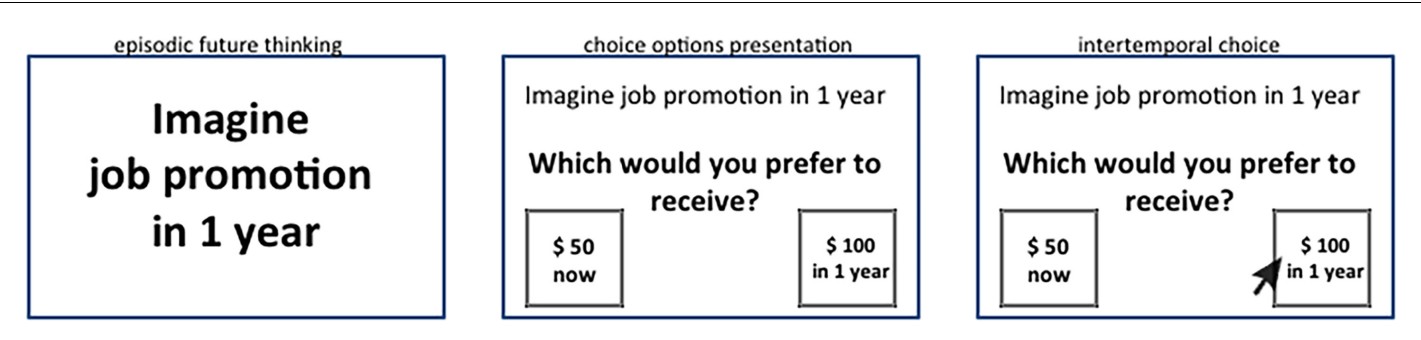

**Figure 2.** Example of an experimental trial in the episodic future thinking (EFT) condition. Participants were presented with an episodic cue and asked to imagine a personal future experience occurring at a specific delay (e.g., in 1 year). They then were presented with two hypothetical reward amounts and indicated their choice between the smaller-immediate reward and the larger-delayed reward to be received at that delay.

discounted large rewards less steeply than small rewards (i.e., magnitude effect; $p < 0.0001$), whereas vmPFC patients discounted large and small rewards at similar rates ($p = 0.76$). Relatedly, vmPFC patients showed steeper DD than controls for the large rewards ($p = 0.04$), but not for the small rewards ($p = 0.67$). Crucially, there was a significant effect of Condition ($F_{1,51} = 54.33$, $p < 0.0001$, partial $\eta^2 = 0.52$), indicating that both healthy controls and vmPFC patients had reduced DD rates in the EFT compared to the Standard condition (EFT effect). There were no other significant effects ($p > 0.07$ in all cases). In particular, the Group × Condition interaction was not significant ($F_{1,51} = 2.18$, $p = 0.14$, partial $\eta^2 = 0.04$).

Although the ANOVA failed to reveal a significant Group × Condition interaction, a limitation of classical statistical analyses like ANOVA is that they do not directly assess the evidence for the null hypothesis, which in this case is that there is no difference in the EFT effect between vmPFC patients and controls. We therefore used a Bayesian approach, which, unlike classical null hypothesis significance testing, can directly compare the evidence for the null hypothesis with the evidence for the alternative hypothesis (*Wagenmakers et al., 2018*). Bayesian-independent samples t-tests were conducted on the EFT effect between vmPFC patients and controls ($AuC_{EFT\ condition} - AuC_{Standard\ condition}$, collapsing across reward magnitudes; vmPFC patients: M = 0.17, SD = 0.21; controls: M = 0.26, SD = 0.17) using the JASP software (*Wagenmakers et al., 2018*). The results show a Bayes factor of 0.738. This value, which compares the likelihood of the alternative hypothesis to the likelihood of the null hypothesis given the present data, represents what *Jeffreys, 1961* termed anecdotal evidence in favor of the null hypothesis. It may be contrasted with the Bayes factor for the group difference in the magnitude effect ($AuC_{Large\ reward} - AuC_{Small\ reward}$, collapsing across the Standard and EFT condition; vmPFC patients: M = 0.01, SD = 0.10; controls: 0.14, SD = 0.13), which equals 11.52, representing strong evidence for the alternative hypothesis and against the null (*Jeffreys, 1961*).

Because previous work has attributed the magnitude effect to processing in the lateral prefrontal cortex (*Ballard et al., 2017*; *Ballard et al., 2018*), we re-ran the previous ANOVA excluding patients with damage touching the lateral prefrontal cortex (N = 4) to assure that they were not driving our results. We confirmed our findings. Again, the ANOVA yielded a Reward magnitude effect ($F_{1,47} = 4.80$, $p = 0.03$, partial $\eta^2 = 0.09$), and, importantly, a Group × Reward magnitude interaction that

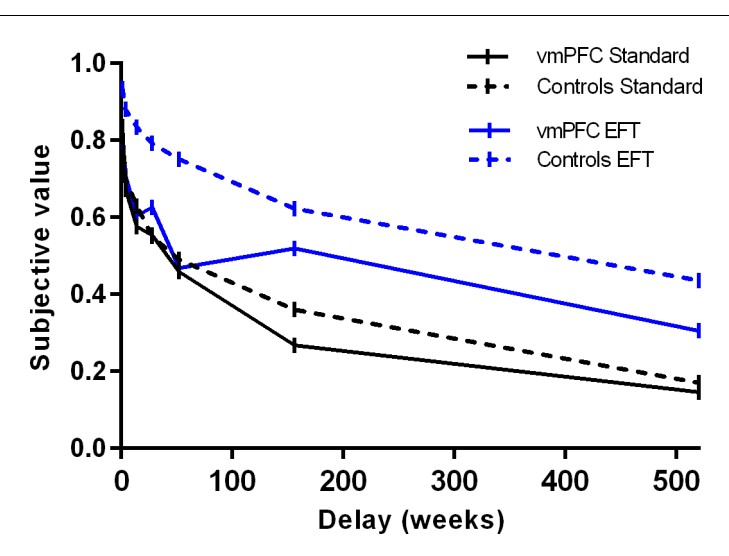

**Figure 3.** Subjective value as a function of delay by participant group and task condition. Lines represent choices averaged across both reward amounts (data points available in the source data file: *Figure 3—source data 1*). The online version of this article includes the following source data and figure supplement(s) for figure 3:

**Source data 1.** Data points for *Figure 3*.

**Figure supplement 1.** Subjective value of small and large rewards as a function of delay for individual participants in the Standard and episodic future thinking (EFT) condition (data points available in the source data file: *Figure 3—figure supplement 1—source data 1*).

**Figure supplement 1—source data 1.** Data points for *Figure 3—figure supplement 1*.

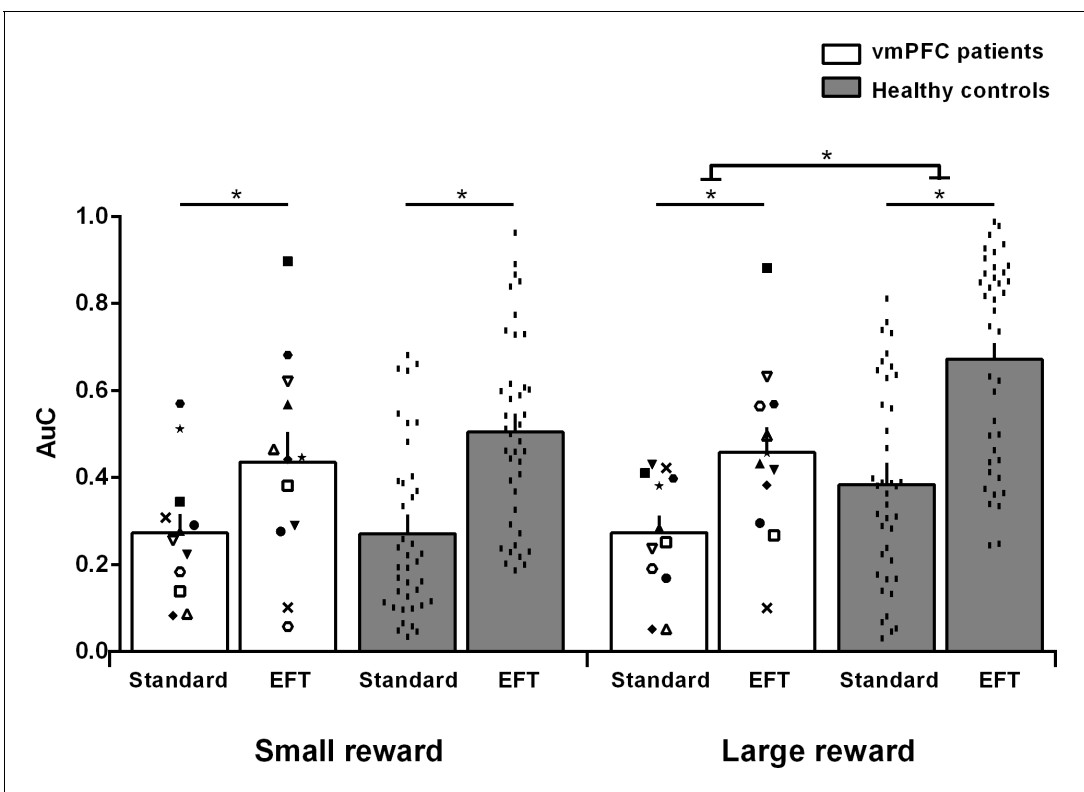

**Figure 4.** Area under the empirical discounting curve (AuC) by participant group, task condition, and reward magnitude. The figure reports individual participants' data. Empty symbols denote ventromedial prefrontal cortex (vmPFC) patients with brain damage touching the lateral prefrontal cortex. *p < 0.05 (data points available in the source data file: *Figure 4—source data 1*).

The online version of this article includes the following source data for figure 4:

**Source data 1.** Data points for *Figure 4*.

was even stronger than in the original ANOVA ($F_{1,47}$ = 13.32, p = 0.0006, partial $\eta^2$ = 0.22), indicating that controls (0.53 vs. 0.39; p < 0.0001), but not vmPFC patients (0.36 vs. 0.40; p = 0.43), discounted large rewards less than small rewards. Note that the magnitude effect was even smaller in patients with damage confined to the vmPFC than in patients with additional damage to the lateral prefrontal cortex (−0.04 vs. 0.10; $t_{(10)}$ = −2.60; p = 0.03), confirming that the lack of a magnitude effect in vmPFC patients was not driven by damage extending beyond vmPFC (*Figure 4*). As in the original ANOVA, there was a main effect of Condition ($F_{1,47}$ = 33.84, p = 0.000001, partial $\eta^2$ = 0.42), and no Group × Condition interaction ($F_{1,47}$ = 2.89, p = 0.095, partial $\eta^2$ = 0.05), confirming reduced DD rates in the EFT compared to the Standard condition in both vmPFC patients and controls. There were no other significant effects (all p's > 0.23).

## Consistency of preference

One possible reason for the poor fit of the hyperbolic function to vmPFC patients' discounting data in the EFT condition is that the data were not monotonically decreasing (*Figure 3*). To directly assess this possibility, we counted the number of 'inconsistent preferences', that is, data points in which the subjective value of a future outcome (amount = R) at a given delay ($R_2$) was greater than that at the preceding delay ($R_1$) by more than 10% of the amount of the future outcome (i.e., $R_2 > R_1 + R/10$, as recommended by *Johnson and Bickel, 2008*; *Sellitto et al., 2010*). An ANOVA on the number of inconsistent preferences with Group, Condition, and Reward magnitude as factors revealed a significant Group × Condition interaction ($F_{1,51}$ = 5.01, p = 0.03, partial $\eta^2$ = 0.09). Post hoc tests indicated that whereas in the Standard condition the number of inconsistent preferences in vmPFC patients did not differ from that of healthy controls (0.75 vs. 0.94; p = 0.31), replicating previous findings (*Sellitto et al., 2010*), in the EFT condition vmPFC patients showed more inconsistent

preferences than controls (1.12 vs. 0.61; p = 0.007). There were no other significant effects (all p's > 0.13).

## Discussion

The present study investigated the effect of vmPFC damage on DD and its responsivity to reward magnitude and cues to imagine personal future events. Three main findings emerged. Whereas healthy controls showed lower DD rates for large compared to small rewards (magnitude effect), vmPFC patients' DD was not modulated by reward magnitude and was abnormally steep for large rewards. By contrast, EFT cues effectively decreased DD in vmPFC patients as well as controls (EFT effect), despite the patients' poor EFT abilities.

Let us first consider the magnitude effect. The tendency to be more likely to choose the delayed option when decisions involve large rewards has been related to self-control mechanisms supported by the lateral prefrontal cortex. This region is more engaged during intertemporal choices between large than small rewards (*Ballard et al., 2017*), and transient disruption of its activity reduces the magnitude effect (*Ballard et al., 2018*). Our finding that the magnitude effect is absent in vmPFC patients points to the vmPFC as another crucial substrate of this effect, and makes contact with previous evidence of impaired sensitivity to magnitude following vmPFC damage (*Peters and D'Esposito, 2016*). These findings support current models of vmPFC as crucially involved in reward valuation during intertemporal choice (*Hare et al., 2009*; *Figner et al., 2010*; *Peters and Büchel, 2011*). We propose that an impaired reward valuation system impeded the (differential) assessment of the utility of large vs. small rewards, interfering with the normal triggering of self-control by the lateral prefrontal cortex for rewards of high perceived value (*di Pellegrino et al., 2007*; *Ballard et al., 2018*).

In the present study, steep DD in vmPFC patients was observed only on choice trials with large rewards, on which greater self-control (shallower discounting) was observed in controls than in patients, consistent with *Peters and D'Esposito, 2016* view that balanced intertemporal choice relies on vmPFC integrity and crosstalk with the lateral prefrontal cortex. We note that previous studies observed steep DD in vmPFC patients even using reward amounts similar in size to our small reward (*Sellitto et al., 2010*; *Peters and D'Esposito, 2016*). The studies, however, had methodological differences from the present effort. Here, we studied DD by sampling delays as long as 10 years, whereas previous studies employed much shorter delays (1 year in *Sellitto et al., 2010*; 60 days in *Peters and D'Esposito, 2016*), which likely changed baseline levels of discounting. The AuC scores of the healthy controls in the present study were indeed lower than those in previous studies. However, the higher baseline rates of discounting for small rewards in the present study, if anything, should have favored the detection of reductions in DD rates with rewards of greater magnitude, and yet no such modulation was observed in vmPFC patients.

Despite being steep at baseline and unresponsive to the amount of reward, vmPFC patients' DD was normally downregulated by cues to imagine the personal future. This finding indicates that vmPFC integrity is not necessary to instill prospection in intertemporal choice with EFT cues. Before discussing this finding further, we note that it rests on accepting the null hypothesis of no group differences in the EFT effect on DD between vmPFC patients and controls. It is unlikely, however, that this null finding simply reflects a lack of statistical power, for example due to a small sample size. First, the null effect on group differences indeed reflects a significant within-participant effect, with greater regard for future amounts in the EFT compared to the Standard condition in vmPFC patients. Second, together with the preservation of the EFT effect, we found a significant reduction of the magnitude effect in the same vmPFC patient sample. Bayesian analyses confirmed greater evidence in favor of the null compared to the alternative hypothesis regarding group differences in the EFT effect on DD.

The finding of a preserved EFT effect on DD in vmPFC patients is surprising in light of previous evidence of impaired EFT in vmPFC patients (*Bertossi et al., 2016a*; *Bertossi et al., 2016b*; *Bertossi et al., 2017*; *Verfaellie et al., 2019*). In healthy individuals, EFT is thought to reduce DD by promoting self-projection into vivid future experiences, boosting the value of future payoffs (*Boyer, 2008*). As expected, the EFT effect is not reliably observed in MTL patients (*Palombo et al., 2015*; *Kwan et al., 2015*), who cannot imagine detail-rich future events (*Race et al., 2011*). Considering that EFT is also heavily compromised in vmPFC patients (*Bertossi et al., 2016a*;

*Bertossi et al., 2016b*; *Bertossi et al., 2017*; *Verfaellie et al., 2019*), how might EFT cues exert influence on their choices?

EFT is supported by a distributed neural network, including vmPFC and the hippocampus (*Schacter et al., 2012*), within which different nodes contribute uniquely to the dynamics of EFT construction. In particular, vmPFC is thought to initiate endogenously the activation of high-level semantic structures (e.g., schemata; *Irish and Piguet, 2013*; *Ghosh et al., 2014*; *McCormick et al., 2017*; *McCormick et al., 2018*), for example, pertaining to the self, one's goals, common events (*D'Argembeau and Mathy, 2011*), around which the hippocampus then builds detail-rich future experiences (*McCormick et al., 2018*; *D'Argembeau, 2020*). Indeed, vmPFC (but not MTL) patients are particularly impaired at imagining self-related (as opposed to other-related) future events, suggesting they fail to activate schematic self-knowledge that favors the collection of individual details for EFT (*Verfaellie et al., 2019*; *D'Argembeau and Mathy, 2011*). Consistent with this idea, recent magnetoencephalography studies show synchronized engagement of vmPFC and the hippocampus during autobiographical memory retrieval and scene construction, with vmPFC activity driving activity in the hippocampus during both the initiation and elaboration of mental events (*Barry et al., 2019*; *McCormick et al., 2020*). Similarly, vmPFC patients are not impaired in constructing future events (*Kurczek et al., 2015*; *Verfaellie et al., 2019*) or scenes (*De Luca et al., 2019*) if the task minimizes the need for self-initiation, whereas the deficit persists in MTL patients (*Kurczek et al., 2015*; *McCormick et al., 2017*; *Verfaellie et al., 2019*). We propose, therefore, that subject-specific event cues, which were self-relevant and familiar to the participants because they had been selected by participants themselves, and were already planned or were plausible in their future, acted as external triggers of self- and situation-relevant schemata, helping to circumvent vmPFC patients' EFT initiation problems. Their intact MTLs allowed them to construct episodic future events, which were then integrated into intertemporal choice, reducing DD. The same benefit would not be expected, and was not found, in patients with severe episodic amnesia due to extensive MTL lesions (*Palombo et al., 2015*; *Kwan et al., 2015*) as their basic deficit in assembling detail-rich experiences cannot be offset by probing semantic structures upstream. An alternative interpretation of the DD modulation is that EFT cues simply shifted attention toward the future, or conferred a positive valence to it, as we encouraged positively valenced EFT. If that were the case, however, one should consistently observe an EFT-induced benefit on DD also in MTL patients, but this is not the case (*Kwan et al., 2015*; *Palombo et al., 2015*).

Functional magnetic resonance imaging (fMRI) evidence has related the EFT effect on DD to the crosstalk between the anterior cingulate cortex (ACC, BA 32) and the hippocampus (*Peters and Büchel, 2010*). Our findings suggest that the ACC is not necessary to update signal values with the EFT output, as this region was lesioned in our vmPFC patients. Our findings are more compatible with the view that in the EFT (vs. Standard) condition, subjective value computation relied on a more distributed network including, in addition to the ACC, the lateral parietal and posterior cingulate cortex (*Peters and Büchel, 2010*). The parietal cortex mediates shifts of attention to memories (*Cabeza et al., 2008*) and across subjective time (*Nyberg et al., 2010*), and the posterior cingulate cortex is implicated in internally directed cognition and EFT (*Schacter et al., 2012*). These regions were found to form a valuation sub-system dedicated to delayed rewards (*Peters and Büchel, 2009*), and may have updated reward value based on EFT, overcoming vmPFC patients' domain-general valuation impairment.

One unexpected finding of our study was that vmPFC patients showed more inconsistent preferences than controls in the EFT condition, while this did not happen in the Standard condition (as in *Sellitto et al., 2010*). One possibility is that vmPFC patients failed at integrating optimally the attributes of choice options with yet another aspect of the choice context, namely, the products of EFT. This interpretation is in line with the role of vmPFC in weighting multiple aspects of choice options (*Pelletier and Fellows, 2019*; *Vaidya et al., 2018*), and in synthetizing the emergent affective quality of a multi-element situation (*Benoit et al., 2014*). It will be important to confirm the unanticipated finding of an association between EFT cueing and inconsistent preferences in vmPFC patients, and to verify whether it extends to other patient populations, as this aspect of DD has not previously been explored (*Kwan et al., 2015*; *Palombo et al., 2015*).

We end by noting some limitations and future directions of our work. Although all 12 patients had lesions centered in the vmPFC, there was some heterogeneity in lesion location, with brain damage extending to the lateral prefrontal cortex in some cases. Our findings held when analyses were

restricted to patients with lesions confined to the vmPFC, but future studies including more vmPFC patients, and a control group of brain-damaged patients, would help confirm the findings and possibly relate them to specific subregions within vmPFC.

In the present study, the order of task conditions was fixed, with the Standard condition always run first, serving as the baseline. Presenting the EFT condition first runs the risk of carryover effects of EFT into the Standard condition, leading to spurious DD baseline levels. This design has been used in previous studies of brain-damaged patients (*Palombo et al., 2015*; *Kwan et al., 2015*), and we deemed it even more suited in vmPFC patients who tend to perseverate. Although the repeated-measures design we chose raises the possibility of practice effects, studies have demonstrated the relative stability of individual discount rates over repeated testing (*Ohmura et al., 2006*; *Harrison and McKay, 2012*).

Finally, our interpretation of vmPFC patients' preserved EFT effect as due to the external cueing of semantic structures driving EFT is speculative at this point. Indeed, this study does not provide direct insight into the type of future representations underlying the EFT effect on DD in vmPFC patients and in healthy controls. For example, although participants were instructed to take 10-15 seconds to think about a personal future event, we did not collect imagination times and thus do not know if patients differed from controls in this regard. We observed that some of the patients may have taken less time than controls to imagine the events, so it remains possible that the quality of EFT in response to the cues differed between groups. Yet, the EFT condition had an effect on DD in both groups. Therefore, the quality and quantity of future event details that are necessary and sufficient to influence DD in vmPFC patients and in healthy controls remain to be investigated. However, our interpretation is consistent with evidence that vmPFC patients produce few internal (episodic) details but a normal number of external (semantic) details during EFT tasks. It is also consistent with current models of vmPFC that postulate this structure is involved in the self-initiation of event construction (e.g., *McCormick et al., 2018*; *Ciaramelli et al., 2019*; *Verfaellie et al., 2019*). Further work should study EFT performance in vmPFC patients under conditions that (externally) promote the selection of self-relevant cues for EFT (as in the present study) vs. those that do not. In this respect, a study by *Kurczek et al., 2015* is worth noting. Unlike previous studies of episodic remembering and EFT (e.g., *Bertossi et al., 2016a*; *Bertossi et al., 2017*), vmPFC patients were guided to choose themselves a specific moment from an extended past or future event to narrate in detail. Under these experimental procedures, vmPFC patients' (re)constructed experience was as context-rich as that of controls, whereas that of MTLs patients remained impoverished nonetheless (*Kurczek et al., 2015*).

To conclude, the present findings reveal different mechanisms governing DD behavior and its flexibility, which differentially rely on vmPFC integrity. In addition, they may inform the clinical assessment and management of impulsivity in patients with vmPFC damage or dysfunction, delineating the boundary conditions for short-sighted choice to emerge, and the contextual manipulations that are or are not expected to push the reach of patients' choice into the future.

## Materials and methods

### Participants

Twelve patients with lesions to vmPFC (8 males; mean age = 57.41 years, SD = 8.20, range = 49–76; mean education = 13.41 years, SD = 3.67; range = 8–20; see *Table 1* for individual patients' demographic and clinical data) and 41 healthy controls (35 males; mean age = 61.09 years, SD = 6.58, range = 49–78; mean education = 13.19 years, SD = 2.82, range = 8–20), matched to patients in age ($t_{51}$ = 1.61; p = 0.11), education ($t_{51}$ = −0.22; p = 0.82), and gender balance ($\chi^2$ = 2.12; p = 0.14), were recruited at the Centre for Studies and Research in Cognitive Neuroscience, Cesena, Italy, and at Baycrest Health Sciences, Toronto, Canada. Patients were selected on the basis of the location of their lesion evident on MRI or computerized tomography (CT) scans (see *Figure 1*) and were tested at least 12 months post-lesion (see Appendix 1 for additional information on patients' recruitment). The lesions of vmPFC patients resulted from rupture of an aneurysm of the anterior communicating artery (in 11 cases) and from stroke of the anterior cerebral artery (in one case). Lesions were bilateral in 10 cases and left-lateralized in the remaining two cases. All participants were screened for any neurological or psychiatric diagnoses likely to affect cognition or interfere with participation.

**Table 1.** vmPFC patients' demographic and clinical data.

| vmPFC patient | Age (y) | Edu (y) | Sex (y) | Time since lesion (y) | EFT Int (z score) | EFT Ext (z score) | PF | LF | DS | LL Imm | LL Del | ROCF copy | ROCF recall |
|---|---|---|---|---|---|---|---|---|---|---|---|---|---|
| P1 (I) | 55 | 13 | M | 4 | −1.42 | 0.58 | 21% | 23% | 34% | 14% | 17% | 100% | 50% |
| P2 (I) | 46 | 13 | M | 7 | −1.54 | −1.44 | 38% | 7% | 49% | 12% | 8% | 100% | 41% |
| P3 (I) | 56 | 8 | M | 13 | −1.43 | −0.73 | 42% | 16% | 23% | 0.43% | 3% | 25% | 2% |
| P4 (I) | 57 | 8 | M | 7 | −1.57 | −0.28 | 42% | 31% | 23% | 7% | 12% | 89% | 27% |
| P5 (C) | 58 | 15 | F | 8 | – | – | 82% | 35% | 18% | 1% | 0.02% | 2% | 13% |
| P6 (C) | 76 | 16 | F | 5 | 0.20 | −0.86 | 55% | 40% | 80% | 81% | 50% | 67% | 62% |
| P7 (C) | 54 | 13 | F | 2 | −2.27 | −1.64 | 58% | 30% | 59% | 2% | 2–3% | 8% | 42% |
| P8 (C) | 65 | 18 | M | 4 | −1.93 | −1.00 | 45% | 2% | 39% | 8% | 7% | 22% | 18% |
| P9 (C) | 56 | 20 | M | 4 | −2.24 | −1.24 | 47% | - | 39% | 1% | 0.7% | 68% | 1% |
| P10 (C) | 51 | 10 | M | 8 | – | – | 45% | 20% | 59% | 4% | 0.7% | 84% | 13% |
| P11 (C) | 66 | 15 | F | 1 | −1.73 | −1.41 | 47% | 55% | 39% | 1% | 1% | 70% | 2% |
| P12 (C) | 49 | 12 | M | 5 | −1.89 | 0.95 | 86% | 50% | 39% | 1% | 0.03% | 58% | 0.7% |

Note: (I) = patient tested in Italy; (C) = patient tested in Canada; M = male; F = female; Edu = education; y = years; vmPFC = ventromedial prefrontal cortex; EFT Int = internal details at the Crovitz episodic future thinking task; EFT Ext = external details at the Crovitz episodic future thinking task; PF = premorbid functioning, based on the full-scale IQ at Wechsler Abbreviated Scale of Intelligence (WAIS–IV; **Wechsler, 2009**) for Canadian patients P7, P9, P12, on the Wechsler test of adult reading (WTAR; **Holdnack, 2001**) for Canadian patients P6 and P11, on the National Adult Reading Test (NART) (**Paolo and Ryan, 1992**) for Canadian patients P5, P8, and P10, and on the Raven Standard Progressive Matrices (SPM) for all Italian patients (**Spinnler and Tognoni, 1987**); LF = letter fluency (**Spinnler and Tognoni, 1987**; **Spreen and Strauss, 1998**); DS = digit span (forward); LL Imm = list learning immediate recall, LL Del = list learning delayed recall, assessed with the Buschke–Fuld Test (**Buschke and Fuld, 1974**; **Spinnler and Tognoni, 1987**) in Italian patients, and with the California Verbal Learning Test-II (**Woods et al., 2006**) in Canadian vmPFC patients; ROCF = Rey–Osterrieth Complex Figure (**Spinnler and Tognoni, 1987**; **Spreen and Strauss, 1998**). For PF, LF, DS, LL, and ROCF, we report percentile scores. Dashes indicate missing data.

They gave informed consent to participate in the study, which was approved by the ethical committees of the University of Bologna, the Regional Health Service of Emilia Romagna, Baycrest Health Sciences, and York University, and in line with the Declaration of Helsinki (*International Committee Of Medical Journal, 1991*).

## Lesion analysis

Individual vmPFC lesions were manually drawn by a highly trained neuroscientist directly on each slice of the normalized T1-weighted template MRI scan from the Montreal Neurological Institute using MRIcro software (*Rorden and Brett, 2000*), based on the most recent MRI or CT scan available. This manual procedure combines segmentation (identification of lesion boundaries) and registration (to a standard template) into a single step, with no additional transformation required (*Kimberg et al., 2007*). Included patients had lesions mainly affecting Brodmann areas (BAs) 10, 11, 32, 24, and 25, with the region of maximal overlap occurring in BAs 11 (M = 12.50 cc, SD = 10.79), 10 (M = 5.70 cc, SD = 6.46), and 32 (M = 3.71 cc, SD = 3.64) (*Figure 1*). Four patients had minimal damage to the lateral prefrontal cortex (BAs 9, 46, 47), but this constituted ~ 5% of their lesion volume, while their vmPFC lesions were on average 10 times larger. Two patients had damage to visual cortex (BAs 17, 18, 19, 37) that constituted ~ 41% and ~ 32% of their lesion volume. These patients did not have visual problems precluding their participation in the study. They attained normal scores on the copy of the Rey–Osterrieth Complex Figure (percentile scores: 66 and 68; *Spreen and Strauss, 1998*) and on the Wechsler Test of Adult Reading (percentile scores: 55 and 47; *Holdnack, 2001*), and proved able to inspect and comprehend a practice trial of the DD task.

## Cognitive profile

The general cognitive functioning of vmPFC patients was preserved in all cases. Patients' performance on standardized tests of executive function and short-term memory was also within the normal range in most cases (mean percentile > 5), whereas long-term memory, as assessed with a list-learning task, was weak in 7 of the 12 patients (see *Table 1* for individual patients'

neuropsychological data). Moreover, most vmPFC patients showed deficits in episodic remembering and EFT, as assessed with the Galton–Crovitz cue-word test, a long-standing method for eliciting autobiographical memories (*Crovitz and Schiffman, 1974*), later adapted to the assessment of EFT (*Addis et al., 2008*; see Appendix 1 for a detailed description of testing procedures in Italy and in Canada). *Table 2* reports the mean number of internal and external details for past and future events produced by vmPFC patients tested in Italy and in Canada and their controls. The results of the Italian patients (a subset of those included in *Bertossi et al., 2016b*) were contrasted with those of the 11 healthy controls from the same study (all males; *Bertossi et al., 2016b*), who were age-matched to the patients (vmPFC patients: M = 47.75, SD = 5.25; healthy controls: M = 41.63, SD = 11.89, $t_{13}$ = −0.97, p = 0.34). The results of the Canadian patients (unpublished) were contrasted with those of 18 healthy controls (10 males; a subset of those included in *Kwan et al., 2016*) age-matched to the patients (vmPFC patients: M = 61.00, SD = 9.83; healthy controls: M = 67.94, SD = 13.57, $t_{22}$ = 1.15, p = 0.26). As for the Italian sample, an ANOVA on the details produced with Group (vmPFC patients, healthy controls), Time (past, future), and Detail (internal, external) as factors showed a significant effect of Time ($F_{1,13}$ = 14.66, p = 0.002, partial $\eta^2$ = 0.53), such that all participants produced more details for past than future events (18.19 vs. 15.37). There were also significant effects of Group ($F_{1,13}$ = 6.16, p = 0.02, partial $\eta^2$ = 0.32) and Detail ($F_{1,13}$ = 9.14, p = 0.009, partial $\eta^2$ = 0.41), qualified by a Group × Detail interaction ($F_{1,13}$ = 8.99, p = 0.01, partial $\eta^2$ = 0.40). Post hoc Fisher tests showed that vmPFC patients produced fewer internal details (11.45 vs. 25.51; p = 0.004) but a similar number of external details than controls (11.39 vs. 11.96; p = 0.89). No other effect was significant (p > 0.31 in all cases). The same ANOVA on the Canadian sample revealed an effect of Group ($F_{1,22}$ = 17.76, p = 0.0003, partial $\eta^2$ = 0.44), qualified by a significant Group × Detail interaction ($F_{1,22}$ = 4.72, p = 0.04, partial $\eta^2$ = 0.18), again indicating that vmPFC patients produced fewer internal details (10.63 vs. 31.78; p = 0.0003) but a similar number of external details than controls (16.79 vs. 25.65; p = 0.09). No other effect was significant (p > 0.32 in all cases). These findings indicate that the previously reported vmPFC patients' impairment in episodic remembering and EFT (*Bertossi et al., 2016b*; *Bertossi et al., 2017*; *Verfaellie et al., 2019*) also applies to the patients tested here.

## DD task

Participants completed a DD task under Standard and EFT conditions. In the Standard condition, over a series of trials, participants viewed pairs of monetary amounts and were asked to make hypothetical choices between an immediate reward and a larger reward available after a delay. For each of two delayed amounts (small magnitude: €80/$100; large magnitude: €1500/$2000), participants were asked to make a block of six choices at each of seven delays (1 week, 1 month, 3 months, 6 months, 1 year, 3 years, and 10 years before receiving the reward), with the resulting 14 blocks pertaining to the different reward amounts and delays presented in random order. Thus, participants made 84 choices in total (2 reward amounts × 7 delays × 6 choices).

In each block, the first choice was between the future amount and half that amount to be received immediately. An iterative, adjusting-amount procedure was used in which the amount of the immediate reward was increased or decreased based on a participant's previous choices, so as to converge on an estimate of the amount of immediate reward that was equivalent in subjective value to the delayed reward (see *Kwan et al., 2015*). The first adjustment was half of the difference between the immediate and delayed amounts presented on the first trial, with each subsequent

**Table 2.** Mean number of internal and external details for past and future events at the Galton–Crovitz cue-word task.

|  | Past events | | Future events | |
|---|---|---|---|---|
|  | Internal details | External details | Internal details | External details |
| Italian vmPFC patients | 13.20 (3.96) | 13.15 (5.74) | 9.69 (0.68) | 9.63 (3.79) |
| Italian healthy controls | 27.84 (10.08) | 12.19 (3.38) | 23.17 (9.05) | 11.73 (4.48) |
| Canadian vmPFC patients | 11.51 (8.07) | 17.88 (8.62) | 9.75 (10.43) | 15.70 (10.21) |
| Canadian healthy controls | 35.34 (9.53) | 26.09 (12.47) | 28.22 (11.24) | 25.21 (10.96) |

Note: Values in parenthesis are standard deviations.

adjustment being half of the preceding adjustment, rounded to the nearest €/$. For example, in the condition where a future reward of $2000 could be received in 3 years, the first choice presented to the participants was '$1000 right now or $2000 in 3 years'. If the participant chose '$2000 in 3 years', the choice on the second trial would be between '$1500 right now' and '$2000 in 3 years'. If the participant then chose '$1500 right now', the choice on the third trial would be '$1250 right now or $2000 in 3 years'. Following the sixth and final trial, the subjective value of the delayed reward was estimated as the amount of the immediate reward that would be presented on a seventh trial. Participants were told that the task assessed preferences, therefore there were no correct or incorrect choices.

The DD task in the EFT condition proceeded as in the Standard condition (i.e., with 2 reward amounts × 7 delays × 6 choices), except that each block of choices was preceded by an EFT cue encouraging participants to imagine vividly a personal future event to occur at that delay (*Figure 2*). In a preliminary session, participants identified planned or plausible personal future events (e.g., appointments, anniversaries, outings) for each of the seven delays in the discounting task. To minimize the possibility of inducing distress, participants were encouraged to include only emotionally neutral or positive future events. If participants encountered difficulties providing an event, the experimenter probed with the following questions: 'Might there be any events with family or friends that may take place in < delay >?" or 'Is there something you could possibly see yourself doing in < delay > or want to do in < delay >?" vmPFC patients had greater difficulty generating events in comparison to controls, and, therefore, all participants were allowed to refer to personal calendars and electronic devices, or to consult with their significant others. Once participants had accessed the relevant event, they described it to the experimenter and labeled it with a short tag. These tags were used as future event cues in the EFT condition. During the cued DD task, upon viewing the EFT cue, participants were instructed to imagine the corresponding personal future event in as much detail as possible, and to press a button when they had had the event in mind for approximately 10–15 s. The button press triggered the decision-making screen, where participants completed intertemporal choices as in the Standard condition. The event cue remained at the top of the screen until the end of the delay block, to reduce demands on memory.

The Standard, uncued version of the task provided a baseline for measuring the effect of future cueing on DD and was run first. The EFT condition was run at least 1 month after the Standard condition. The experimental conditions were administered in this fashion to avoid carryover effects of the EFT condition, which would likely contaminate the baseline condition (for a similar approach, see *Palombo et al., 2015*; *Kwan et al., 2015*). A growing body of research indicates EFT is an effective strategy to reduce DD (reviewed in *Rung and Madden, 2018*; *Bulley and Schacter, 2020*), and, as such, it is expected to have carryover effects. Therefore, participants undergoing an EFT condition first might continue to engage in EFT while making choices in the following Standard condition, especially vmPFC patients who are subject to perseveration.

## Assessment of DD rates

Preliminary fits of the hyperbolic function SV = 1/(1+kD), with SV = subjective value (expressed as a fraction of the delayed amount) and D = delay (in days), to individual participants' data using a non-linear least-squares algorithm (implemented in Statistica Statsoft) revealed that subjective preferences were not equally well characterized by hyperbolic functions in the Standard and EFT conditions, especially in vmPFC patients, whose discounting curves in the EFT condition were not always monotonically decreasing (see *Figure 3* and *Figure 3—figure supplement 1*). An ANOVA on $R^2$ values with Group (vmPFC patients, healthy controls), Condition (Standard, EFT), and Reward magnitude (small, large) as factors confirmed a significant effect of Condition ($F_{1,51}$ = 7.20, p = 0.009, partial $\eta^2$ = 0.12) reflecting the fact that $R^2$ values were lower in the EFT condition than in the Standard condition (0.54 vs. 0.64). The Group × Condition interaction, which just failed to reach statistical significance ($F_{1,51}$ = 3.99, p = 0.050, partial $\eta^2$ = 0.07), suggests that the effect of Condition was driven by vmPFC patients (healthy controls: 0.62 vs. 0.58; vmPFC patients: 0.72 vs. 0.41). No other effects were significant (all p's > 0.09).

Given that subjective preferences were not equally well characterized by hyperbolic functions in the Standard and EFT conditions across groups, the degree to which participants discounted delayed rewards (DD rate) was quantified using the AuC, a measure of DD that does not depend on theoretical assumptions on the shape of the discounting curve (*Myerson et al., 2001*). Delays and

subjective values were normalized. Each delay was expressed as a proportion of maximum delay (120 months) and subjective values were expressed as a proportion of the delayed values. The normalized delays were then plotted on the x axis and the normalized subjective values on the y axis as a function of delay to construct a discounting curve. Vertical lines were drawn from each x value to the curve, subdividing the AuC into a series of trapezoids. The area of each trapezoid was calculated as $(x_2 - x_1)(y_1 + y_2)/2$, where $x_1$ and $x_2$ are successive delays, and $y_1$ and $y_2$ are the subjective values associated with those delays (*Myerson et al., 2001*). The AuC is the sum of the areas of all the trapezoids. The AuC varies between 0 (maximally steep discounting) and 1 (no discounting). The smaller the AuC, the steeper the DD, and the more participants were inclined to choose smaller-immediate rewards over larger-delayed rewards.

## Statistical analyses

Measures of interest were entered in repeated-measures ANOVAs with Group (vmPFC patients, healthy controls) as the between-subject factor and Condition (Standard, EFT) and Reward magnitude (small, large) as within-subject factors (see *Source data 1*). Post hoc analyses were conducted with the Fisher test. We report results significant at $p < 0.05$, two-tailed, and partial $\eta^2$ as measure of effect size.

## Acknowledgements

The authors acknowledge the support of a Progetti di Rilevante Interesse Nazionale (PRIN) grant from the Italian Ministry of Education, University, and Research (#20174TPEFJ) to EC, a York Research Chair and the Canada First Research Excellence Fund to RSR (CFREF-2015-00013), and a grant from the National Institute on Aging of the National Institutes of Health (# R01AG058885) to LG and JM. The authors thank Jake Kurczek and Manuela Sellitto for their suggestions on the paper.

## Additional information

### Funding

| Funder | Grant reference number | Author |
| --- | --- | --- |
| Ministry of Education, University and Research | 20174TPEFJ | Elisa Ciaramelli |
| National Institute on Aging | R01AG058885 | Leonard Green Joel Myerson |
| Canada First Research Excellence Fund | CFREF-2015-00013 | R Shayna Rosenbaum |

The funders had no role in study design, data collection and interpretation, or the decision to submit the work for publication.

### Author contributions

Elisa Ciaramelli, Conceptualization, Formal analysis, Supervision, Methodology, Writing - original draft, Writing - review and editing; Flavia De Luca, Conceptualization, Data curation, Formal analysis, Writing - original draft, Writing - review and editing; Donna Kwan, Conceptualization, Data curation, Software; Jenkin Mok, Data curation, Investigation; Francesca Bianconi, Violetta Knyagnytska, Data curation, Formal analysis; Carl Craver, Leonard Green, Joel Myerson, Conceptualization, Investigation, Writing - review and editing; R Shayna Rosenbaum, Conceptualization, Data curation, Software, Formal analysis, Supervision, Investigation, Methodology, Writing - review and editing

### Author ORCIDs

Elisa Ciaramelli (iD) https://orcid.org/0000-0002-4797-8060

### Ethics

Human subjects: Participants gave informed consent to participate in the study, which was approved by the ethical committees of the University of Bologna, the Regional Health Service of Emilia

Romagna (#1586), Baycrest Health Sciences (#08-57), and York University, and in line with the Declaration of Helsinki (International Committee of Medical Journal Editors, 1991).

## Decision letter and Author response

Decision letter https://doi.org/10.7554/eLife.67387.sa1
Author response https://doi.org/10.7554/eLife.67387.sa2

## Additional files

### Supplementary files

- Source data 1. Statistical analyses.
- Transparent reporting form

### Data availability

Data that support the findings of this study are available at https://doi.org/10.5061/dryad.w3r2280qf.

The following dataset was generated:

| Author(s) | Year | Dataset title | Dataset URL | Database and Identifier |
|-----------|------|---------------|-------------|-------------------------|
| Ciaramelli E | 2021 | Data from: The role of ventromedial prefrontal cortex in reward valuation and future thinking during intertemporal choice | https://doi.org/10.5061/dryad.w3r2280qf | Dryad Digital Repository, 10.5061/dryad.w3r2280qf |

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

## Appendix 1

### Materials and methods

#### Patient recruitment

Patients were recruited at the Centre for Studies and Research in Cognitive Neuroscience, Cesena, Italy, and at Baycrest Health Sciences, Toronto, Canada, between 2015 and 2019. Patients with relatively restricted lesions to vmPFC are rare, and there are no previous studies on the effect of EFT cueing on DD in these patients. Thus, the number of participants was based on previous studies of DD in vmPFC patients (e.g., *Sellitto et al., 2010*: 7 vmPFC patients, 20 healthy controls; *Peters and D'Esposito, 2016*: 9 vmPFC patients, 19 healthy controls; *Fellows and Farah, 2005*: 12 vmPFC patients, 26 healthy controls). A somewhat larger N (= 41) for healthy participants was chosen based on previous behavioral findings where the influence of EFT on DD was found using a group of 30 healthy adults (*Peters and Büchel, 2010*).

One of the four Italian patients included in the study had participated in a previous study on uncued DD (*Sellitto et al., 2010*). All eight Canadian patients had taken part in study on DD and probability discounting (both without cues) conducted shortly before the present experiment (*Mok et al., 2021*), and their uncued DD data are included in the current Standard condition data. As for EFT, all Italian patients had participated in two EFT studies run between 2013 and 2015 (*Bertossi et al., 2016b*; *Bertossi et al., 2017*), whereas all Canadian patients were tested between 2015 and 2019, with results reported for the first time here (see *Table 1* and *Table 2*).

#### Assessment of EFT

The Galton–Crovitz cue-word test is a long-standing method for eliciting autobiographical memories (*Crovitz and Schiffman, 1974*), later adapted to the assessment of EFT (*Addis et al., 2008*). The same general testing and scoring procedures for the cue-word test were followed in the Italian and Canadian laboratories, with minor differences. Participants were presented with cue words (9/6 cues per condition in Italy/Canada) and were asked to remember past personal events (up to 5 years ago) and to imagine future personal events (up to 5 years into the future). For 'past' trials, participants were asked to recall personally experienced events at specific times and places. For 'future' trials, participants were asked to imagine specific novel events that they might experience in the future. Remembered/imagined events were to last minutes or hours but not more than a day. Participants recounted the event they had in mind for 3/5 min (in Italy/Canada), followed by a general probe encouraging greater usage of details ('Is there anything else you can remember/imagine?'). Narratives were scored using the Autobiographical Interview (see *Levine et al., 2002*; *Addis et al., 2008*): for each event, narratives were segmented into distinct details, which were categorized as either internal (referring to specific episodic information about the central event) or external (e.g., semantic information, information unrelated to the central event, metacognitive/editorializing statements). Internal and external details were tallied and averaged across trials.

### Results

#### DD rates-control analysis on cultural effects

We tested whether there were cross-cultural differences in the EFT-driven modulation of DD. We repeated our main ANOVA on AuCs with Group (vmPFC patients, healthy controls), Condition (Standard, EFT), and Reward magnitude (small, large) as factors, this type considering Testing site (Italy, Canada) as an additional factor. We confirmed our findings, which held across different testing sites. Again, the ANOVA yielded a Reward magnitude effect ($F_{1,49}$ = 12.93, p = 0.0007, partial $\eta^2$ = 0.20) and a Group × Reward magnitude interaction ($F_{1,49}$ = 7.97, p = 0.006, partial $\eta^2$ = 0.13), indicating that controls (0.53 vs. 0.39; p = 0.001), but not vmPFC patients (0.37 vs. 0.35; p = 0.70), discounted large rewards less than small rewards. Moreover, there was a main effect of Condition ($F_{1,49}$ = 45.84, p = 0.00001, partial $\eta^2$ = 0.48), confirming reduced DD rates in the EFT compared to the Standard condition in both vmPFC patients and controls. The Group × Condition interaction was not significant ($F_{1,49}$ = 1.44, p = 0.23, partial $\eta^2$ = 0.03). There were no other significant effects (p > 0.11 in all cases) and, in particular, testing site had no effect and did not figure in any significant interaction (p > 0.14 in all cases).

