## [Decision Letter]

**Acceptance summary:**

This paper provides important data on the contribution of the vmPFC to temporal discounting, showing that vmPFC lesions alter the effect of reward magnitude but not episodic future thinking on delay discounting. This finding will be of interest to cognitive neuroscientists working on prospection, decision-making and executive control.

**Decision letter after peer review:**

Thank you for submitting your article "The role of ventromedial prefrontal cortex in reward valuation and future thinking during intertemporal choice" for consideration by *eLife*. Your article has been reviewed by 3 peer reviewers, and the evaluation has been overseen by a Reviewing Editor and Michael Frank as the Senior Editor. The following individual involved in review of your submission has agreed to reveal their identity: Jan Peters (Reviewer #2).

Essential Revisions:

1. All reviewers agreed that your data showing an effect of VMPFC on magnitude but not EFT are important as they are counter to popular theories about VMPFC function.

However, all reviewers found it difficult to reconcile your stated hypothesis with the study design. The last paragraph of the introduction mentions a positive prediction of the EFT effect based on previous work by Bertossi et al. but also a null prediction based on Ghosh et al. The paragraph is written in a confusing way as it ends by stating that you predict the null result without mentioning the alternative. It would be important to clearly state your hypothesis and how it informed the experimental design.

In this regard, reviewers noted that the experiment does not allow testing whether cues would circumvent potential schema activation deficits that would result in a null effect for the EFT manipulation. It seems a different set of experiments would be required to test this question.

Instead, the experiment offers a perfectly reasonable way to test the hypothesis that VMPFC is important for both the EFT and magnitude manipulation. Your results show that only the magnitude but not EFT effect was impaired in the VMPFC group, which all reviewers agreed is an interesting and potentially important finding. This hypothesis would explain the absence of a test of the mechanism and the speculative discussion about the reason for the null observation.

We ask that you clarify the hypothesis for conducting this study, and clearly distinguish between a priori hypothesis and post-hoc explanations of the findings throughout the manuscript.

2. Reviewers made a number of additional comments and suggestions which we would ask you to consider when preparing a revised manuscript. In particular, the inconsistency analysis and the Autobiographical Interview data need clarification.

*eLife*'s editorial process also produces an assessment by peers designed to be posted alongside a preprint for the benefit of readers.

*Reviewer #1 (Recommendations for the authors):*

1. Within the main text (discussion), please discuss issues with accepting the null hypothesis, and point out that interpretation of the current findings need to consider this potential caveat.

2. Unless the authors can better explain their reasoning for the strong hypothesis of no group differences regarding the EFT effects, maybe the introduction could be written in a more balanced manner regarding the potential outcomes. This would be more aligned with the discussion where the external cueing of semantic structures is presented as a speculative interpretation, rather than a confirmation of the hypothesis.

3. Were any of the choices realized and paid out? If not, where subjects informed these were hypothetical choices? This should be clarified in the manuscript.

4. Please provide source data files (spreadsheets) with individual data points for figures 3 and 4, as well as Supplementary figure 1.

5. Please upload the lesion map shown in Figure 1 as nii file to a public repository (e.g., neurovault.org).

*Reviewer #2 (Recommendations for the authors):*

1. The authors note that their interpretation of the preserved EFT effects in the vmPFC patients in terms of e.g. semantic processing remains speculative, but is supported by the finding of intact external details production following vmPFC damage in earlier studies. But was this also the case in the present data set? This remains unclear, because for the AI data, only z-scores relative to some earlier control group (Kwan et al. 2015) are reported (Table 1 and Supplement p. 30). Was this control group matched to the patients? And since the referenced Kwan et al. (2015) paper reports only on six patients (presumably the patients from the Canada site?) – what about the patients from the Italian site, which control group were their AI data compared to?

2. Directly related to my previous point: The methods section states that external details were in the normal range in the vmPFC group (mean z-score for EFT = -.73) but from Table 1 we can see that 8/10 patients in fact exhibit a negative z-score. This suggests that a direct group comparison of the external details scores would very likely reveal a significant group difference. Generally, it would help to report to actual control data here, not just the z-scores, and report the respective group comparisons.

3. The description of the inconsistency analysis was somewhat unclear. The authors use the procedure suggested by Johnson and Bickel (2008), which makes sense, given the overall analytical approach that focuses on the analysis of indifference points. However, this procedure is based on a comparison of adjacent indifference points. In contrast, the authors are referring to the number of inconsistent choices – this is either a typo, or a different procedure. I think the former, because the reported absolute numbers (e.g. means around 1) and the single subject plots in the supplement appear to reflect the number of inconsistent ID points rather than choices. If this is the case, I disagree with the statement that the "mean number of inconsistent choices was very low" (p. 10) – as this probably reflects the mean number of inconsistent indifference points and not choices, about 1 out of 6 ID points was inconsistent in the vmPFC group, which is a lot.

4. The EFT cues are suggested to help vmPFC patients to "circumvent their initiation problems" (p. 12) but I am not sure I follow this logic. First, the AI procedure typically entails external cues as well, and here vmPFC patients showed impairments (Table 1, but see my point 1 above). Second, some of the cited papers (e.g. Verfaellie et al., 2019) also used specific event cues, and still observed reduced internal details production in vmPFC patients.

5. One shortcoming with the paper is that no data are available that could inform *how* vmPFC patients might have utilized the EFT cues, and whether the processes at work might have differed from those in controls. Many points mentioned in the discussion (self-referential processing, semantic processing, activation of schemata, self-initiation vs. external cueing etc.) thus necessarily remain conjecture.

– Generally, the EFT analysis might benefit from a Bayesian analysis, given that one conclusion from the EFT data is a null effect (absence of evidence a for group difference in EFT modulation).

*Reviewer #3 (Recommendations for the authors):*

1. I had a hard time understanding the rationale for the authors' prediction that the EFT effect would be preserved in vmPFC patients, which seems to be the main new result of this manuscript. Several past studies have found that patients with vmPFC damage have impairments in constructing scenes and imagining future events (e.g. Bertossi et al., 2016; De Luca et al., 2018; Fellows and Farah, 2005), and even the participants in the current study provided less internal details on the Crovitz interview – indicating a deficit in EFT. I would thus have expected a reduction in the influence of the EFT manipulation on the discounting behavior of vmPFC patients – assuming that this effect is related to participants' ability to vividly imagine future events. The authors instead argue that because the EFT cues in this task referenced personal events, they would act to externally activate EFT, thus circumventing this deficit and leading to preserved performance. This seems like a very unlikely hypothesis to form a priori, and raises the question of why the authors would design the task in such a way where they would make it harder to distinguish this hypothesis from the null hypothesis that the EFT effect is preserved in these patients regardless of the cues used. I am particularly worried that this prediction was made post hoc to satisfy the observed pattern of the data.

2. Relatedly, the discounting behavior of vmPFC patients in the EFT condition was more internally inconsistent than control participants, making it harder to satisfactorily argue that this was a solidly null effect. I am also curious if the staircasing procedure used here to estimate participants' SV for different offers was sufficient to reveal the preferences of these patients. vmPFC patients also tend to be more internally inconsistent in repeated preference-based choices (e.g. Yu, Dana and Kable, bioRxiv), and so it seems possible that the six repeated choices may have produced less stable preferences and hence noisier estimates of discounting behavior. It would be helpful at least as a control analysis if the authors could show that this procedure was similarly successful in converging on preferences for both patients and controls.

3. Did performance of vmPFC damaged patients on the Crovitz interview differ for past versus future events? This could be revealing as to the role of the vmPFC in recalling previous episodes in detail versus constructing future or imagined scenes. Theories about the role of vmPFC and hippocampus in scene construction link these processes together, and so it would be helpful to test whether both are impaired together here or if these processes might be at least somewhat dissociable. Relatedly, can the authors be certain that the reduction of internal details in the Crovitz interview reflect deficits in the construction of imagined events? For example, could these deficits also be related to reductions in semantic fluency or narrative skills?

4. The authors test if there is a relation between EFT based on performance in the Crovitz interview and the influence of the EFT manipulation on discounting in the 12 vmPFC patients. This analysis is likely very underpowered to detect a significant result and thus hard to interpret. I would suggest removing this analysis from the manuscript.

5. The pattern of lesions in this sample is fairly widespread and includes several areas outside of the frontal lobes, including the cerebellum, thalamus and occipital lobe in at least one patient (though not necessarily the same patient). While this kind of widespread damage outside the ROI is a common and expected problem in a human lesion study, the inclusion of a control group of patients with damage outside of vmPFC would have provided more confidence in these results being related to vmPFC damage more specifically, and not other non-specific factors confounded with brain damage (e.g. psychoactive medication use). I do not believe that the authors necessarily need to recruit a lesion control sample, but I feel that this limitation should be addressed in the discussion.

[Editors' note: further revisions were suggested prior to acceptance, as described below.]

Thank you for resubmitting your work entitled "The role of ventromedial prefrontal cortex in reward valuation and future thinking during intertemporal choice" for further consideration by *eLife*. Your revised article has been evaluated by Michael Frank (Senior Editor) and a Reviewing Editor.

The manuscript has been improved but there is one remaining issue that needs to be addressed, as outlined below:

Responses to EFT cues were self-paced and subjects were instructed to respond within 10-15 seconds. It seems important to know whether there were any group differences in the time spent with the EFT cues. For instance, if patients took more time, this could indicate an actual impairment in EFT that failed to affect choices because it was compensated by taking more time. Please include an analysis of the responses time data in the manuscript, and, in case there are significant differences, discuss these effects.

---

## [Author Response]

Essential Revisions:1. All reviewers agreed that your data showing an effect of VMPFC on magnitude but not EFT are important as they are counter to popular theories about VMPFC function.However, all reviewers found it difficult to reconcile your stated hypothesis with the study design. The last paragraph of the introduction mentions a positive prediction of the EFT effect based on previous work by Bertossi et al. but also a null prediction based on Ghosh et al. The paragraph is written in a confusing way as it ends by stating that you predict the null result without mentioning the alternative. It would be important to clearly state your hypothesis and how it informed the experimental design.In this regard, reviewers noted that the experiment does not allow testing whether cues would circumvent potential schema activation deficits that would result in a null effect for the EFT manipulation. It seems a different set of experiments would be required to test this question.Instead, the experiment offers a perfectly reasonable way to test the hypothesis that VMPFC is important for both the EFT and magnitude manipulation. Your results show that only the magnitude but not EFT effect was impaired in the VMPFC group, which all reviewers agreed is an interesting and potentially important finding. This hypothesis would explain the absence of a test of the mechanism and the speculative discussion about the reason for the null observation.We ask that you clarify the hypothesis for conducting this study, and clearly distinguish between a priori hypothesis and post-hoc explanations of the findings throughout the manuscript.

We understand the reviewers’ point, and we agree that this paradigm does not allow testing whether subject-specific personally relevant cues, such as those used in our study, are indeed effective in externally initiating EFT in vmPFC patients. We concur that, for the sake of clarity, this is best presented as a speculative discussion of the preserved EFT effect on DD in vmPFC patients (based on Ghosh et al., 2014), and not among the a priori hypotheses. In the Introduction, we now only formulate the hypothesis based on previous evidence of impaired EFT in vmPFC patients (e.g., Bertossi et al., 2016, etc.), which would lead to the prediction of a reduced EFT effect in vmPFC patients.

Thus, p. 5 now reads: “Concerning prospection, previous studies have observed an EFT effect on DD, such that people discount future rewards less steeply if cued to imagine personal future events during intertemporal choice (Peters and Büchel, 2010; Benoit et al., 2011). Considering that vmPFC is implicated in prospection (Schacter et al., 2012), and that vmPFC patients are impaired in EFT (Bertossi et al., 2016a,b; Bertossi et al., 2017), vmPFC patients' DD should remain steep even when EFT cues are provided, because patients may nevertheless fail to construct the vivid future events that might be needed to counteract DD. Thus, we predict a reduced EFT effect on DD in vmPFC patients compared to healthy controls.”

2. Reviewers made a number of additional comments and suggestions which we would ask you to consider when preparing a revised manuscript. In particular, the inconsistency analysis and the Autobiographical Interview data need clarification.eLife's editorial process also produces an assessment by peers designed to be posted alongside a preprint for the benefit of readers.

We have now clarified the analysis on the inconsistent preference. This pertains to indifference points, not single choices (as in our previous study; Sellitto et al., 2010), as Reviewer 2 correctly inferred. Inconsistent preferences are defined as intended on p. 8 as “data points in which the subjective value of a future outcome (amount = R) at a given delay (R_2_) was greater than that at the preceding delay (R_1_) by more than 10% of the amount of the future outcome (i.e., R_2_ > R_1_ + R/10, as in Sellitto et al., 2010)”; to avoid confusion, we have now corrected the expression ‘inconsistent choice’ to ‘inconsistent preference’ throughout the paper.

As for the Autobiographical (Crovitz) task, as suggested by Reviewer 2, we now directly contrast internal and external details produced at the Crovitz task by vmPFC patients tested in Italy and in Canada with their respective control groups. As the two ANOVAs show, we confirm that, relative to controls, both patient groups produced fewer internal (episodic) details but a similar number of external (semantic) details during EFT (as well as episodic remembering). Therefore, the previously reported EFT problems in vmPFC patients also apply to the patients tested here.

Pp. 16-17 now read: “Table 2 reports the mean number of internal and external details for past and future events produced by vmPFC patients tested in Italy and in Canada and their controls. The results of the Italian patients (a subset of those included in Bertossi et al. 2016b) were contrasted with those of the 11 healthy controls from the same study (all males; Bertossi et al., 2016b) who were age-matched to the patients (vmPFC patients: M = 47.75, SD = 5.25; healthy controls: M = 41.63, SD = 11.89, t_13_ = -0.97, p = 0.34). The results of the Canadian patients (unpublished) were contrasted with those of 18 healthy controls (10 males; a subset of those included in Kwan et al., 2015) age-matched to the patients (vmPFC patients: M = 61.00, SD = 9.83; healthy controls: M = 67.94, SD = 13.57, t_22_ = 1.15, p = 0.26). As for the Italian sample, an ANOVA on the details produced with Group (vmPFC patients, healthy controls), Time (Past, Future), and Detail (internal, external) as factors showed a significant effect of Time (F_1,13_ = 14.66, p = 0.002, partial η^2^ = 0.53), such that all participants produced more details for past than future events (18.19 vs. 15.37). There were also significant effects of Group (F_1,13_ = 6.16, p = 0.02, partial η^2^ = 0.32) and of Detail (F_1,13_ = 9.14, p = 0.009, partial η^2^ = 0.41), qualified by a Group x Detail interaction (F_1,13_ = 8.99, p = 0.01, partial η^2^ = 0.40). Post hoc Fisher tests showed that vmPFC patients produced fewer internal details (11.45 vs. 25.51; p = 0.004) but a similar number of external details than controls (11.39 vs. 11.96; p = 0.89). No other effect was significant (p > 0.31 in all cases). The same ANOVA on the Canadian sample revealed an effect of Group (F_1,22_ = 17.76, p = 0.0003, partial η^2^ = 0.44), qualified by a significant Group x Detail interaction (F_1,22_ = 4.72, p = 0.04, partial η^2^ = 0.18), again indicating that vmPFC patients produced fewer internal details (10.63 vs. 31.78; p = 0.0003) but a similar number of external details than controls (16.79 vs. 25.65; p = 0.09). No other effect was significant (p > 0.32 in all cases).

Reviewer #1 (Recommendations for the authors):1. Within the main text (discussion), please discuss issues with accepting the null hypothesis, and point out that interpretation of the current findings need to consider this potential caveat.

We have added a discussion of this caveat on p. 10, which reads: “Before discussing this finding further, we note that it rests on accepting the null hypothesis of no group differences in the EFT effect on DD between vmPFC patients and controls. It is unlikely, however, that this null finding simply reflects a lack of statistical power, for example due to a small sample size. First, the null effect on group differences indeed reflects a significant within-participant effect, with greater regard for future amounts in the EFT compared to the Standard condition in vmPFC patients. Second, together with the preservation of the EFT effect, we found a significant reduction of the magnitude effect in the same vmPFC patient sample. Bayesian analyses confirmed greater evidence in favour of the null compared to the alternative hypothesis regarding group differences in the EFT effect on DD.”

2. Unless the authors can better explain their reasoning for the strong hypothesis of no group differences regarding the EFT effects, maybe the introduction could be written in a more balanced manner regarding the potential outcomes. This would be more aligned with the discussion where the external cueing of semantic structures is presented as a speculative interpretation, rather than a confirmation of the hypothesis.

We agree and thank the reviewer for this suggestion. See p. 5 and our response to Essential revisions point #1 above.

3. Were any of the choices realized and paid out? If not, where subjects informed these were hypothetical choices? This should be clarified in the manuscript.

These were hypothetical choices, and none was paid out. Participants knew these were hypothetical choices. We now clarify this on p. 18 and in the Abstract. Please note that in a previous study from our group (Sellitto et al., 2010), we conducted a corollary investigation of DD for money in vmPFC patients and healthy controls using real rewards. We obtained similar findings, and vmPFC patients still discounted real monetary rewards more steeply thancontrols.

4. Please provide source data files (spreadsheets) with individual data points for figures 3 and 4, as well as Supplementary figure 1.

We have added source data files for Figure 3, Figure 4, and Figure 3—figure supplement 1.

5. Please upload the lesion map shown in Figure 1 as nii file to a public repository (e.g., neurovault.org).

We have added the lesion nii files (individual patients’ lesions and overlap) to the raw data file, accessible at Dryad.

Reviewer #2 (Recommendations for the authors):– Generally, the EFT analysis might benefit from a Bayesian analysis, given that one conclusion from the EFT data is a null effect (absence of evidence a for group difference in EFT modulation).

We thank the reviewer for this suggestion. We have added a Bayesian analysis of group differences in the EFT effect, described on p. 7 as follows: Although the ANOVA failed to reveal a significant Group x Condition interaction, a limitation of classical statistical analyses like ANOVA is that they do not directly assess the evidence for the null hypothesis, which in this case is that there is no difference in the EFT effect between vmPFC patients and controls. We therefore used a Bayesian approach, which, unlike classical null hypothesis significance testing, can directly compare evidence for the null hypothesis with evidence for the alternative hypothesis (Wagenmakers et al., 2018). Bayesian independent samples t-tests were conducted on the EFT effect between vmPFC patients and controls (AuC_EFT condition_ – AuC_Standard condition_, collapsing across reward magnitudes; vmPFC patients: M = 0.17, SD = 0.21; controls: M = 0.26, SD = 0.17) using the JASP software (Wagenmakers et al., 2018). The results show a Bayes factor of 0.738. This value, which compares the likelihood of the alternative hypothesis to the likelihood of the null hypothesis given the present data, represents what Jeffreys (1961) termed anecdotal evidence in favor of the null hypothesis. It may be contrasted with the Bayes factor for the group difference in the magnitude effect (AuC_Large reward_ – AUC_Small reward_, collapsing across the Standard and EFT condition; vmPFC patients: M = 0.01, SD = 0.10; controls: 0.14, SD = 0.13), which equals 11.52, representing strong evidence for the alternative hypothesis and against the null (Jeffreys, 1961).”

Reviewer #3 (Recommendations for the authors):1. I had a hard time understanding the rationale for the authors' prediction that the EFT effect would be preserved in vmPFC patients, which seems to be the main new result of this manuscript. Several past studies have found that patients with vmPFC damage have impairments in constructing scenes and imagining future events (e.g. Bertossi et al., 2016; De Luca et al., 2018; Fellows and Farah, 2005), and even the participants in the current study provided less internal details on the Crovitz interview – indicating a deficit in EFT. I would thus have expected a reduction in the influence of the EFT manipulation on the discounting behavior of vmPFC patients – assuming that this effect is related to participants' ability to vividly imagine future events. The authors instead argue that because the EFT cues in this task referenced personal events, they would act to externally activate EFT, thus circumventing this deficit and leading to preserved performance. This seems like a very unlikely hypothesis to form a priori, and raises the question of why the authors would design the task in such a way where they would make it harder to distinguish this hypothesis from the null hypothesis that the EFT effect is preserved in these patients regardless of the cues used. I am particularly worried that this prediction was made post hoc to satisfy the observed pattern of the data.

We agree with the reviewer. We have now reframed our hypotheses as suggested by the reviewers and the editors. In the Introduction, we now make only the hypothesis of a reduced EFT effect on DD in vmPFC patients, which is based on previous evidence of an EFT impairment in vmPFC patients (Bertossi et al., 2016a,b; 2017). We now reserve the alternative interpretation that vmPFC is critical for schema instantiation for the Discussion section, as it is reinforced by the findings, which we did not initially anticipate (see p.5 and our response to point #1 above).

2. Relatedly, the discounting behavior of vmPFC patients in the EFT condition was more internally inconsistent than control participants, making it harder to satisfactorily argue that this was a solidly null effect. I am also curious if the staircasing procedure used here to estimate participants' SV for different offers was sufficient to reveal the preferences of these patients. vmPFC patients also tend to be more internally inconsistent in repeated preference-based choices (e.g. Yu, Dana and Kable, bioRxiv), and so it seems possible that the six repeated choices may have produced less stable preferences and hence noisier estimates of discounting behavior. It would be helpful at least as a control analysis if the authors could show that this procedure was similarly successful in converging on preferences for both patients and controls.

It is difficult to determine whether patients showed stable (single) choices in the DD task, because there are not repeated choices. So, if the patient selects the immediate option in one trial (e.g., $20 now vs. $40 in 10 years) and the delayed option the following trial (e.g., $10 $ now vs. $40 in 10 years), it is impossible to know whether they inconsistently switched from displaying impulsivity to displaying patience or if they did so because the amounts had changed. To answer the reviewer’s question, it would perhaps be necessary to administer the same DD task repeatedly. In the past, we have tried to administer the DD task repeatedly to vmPFC patients; we did not look at the consistency of single/specific choices, but the patients were consistently impulsive. We have no way to re-access the set of patients included in our study at this time. Please note, however, that in the Standard condition, patients’ preferences were not inconsistent, and their subjective value function obeyed a hyperbolical function similar to that seen in controls, so it does not seem that the staircase procedure was generally unable to capture vmPFC patients’ preferences; rather, EFT increased inconsistent preferences in vmPFC patients, a finding that we discuss on pp. 12-13.

3. Did performance of vmPFC damaged patients on the Crovitz interview differ for past versus future events? This could be revealing as to the role of the vmPFC in recalling previous episodes in detail versus constructing future or imagined scenes. Theories about the role of vmPFC and hippocampus in scene construction link these processes together, and so it would be helpful to test whether both are impaired together here or if these processes might be at least somewhat dissociable. Relatedly, can the authors be certain that the reduction of internal details in the Crovitz interview reflect deficits in the construction of imagined events? For example, could these deficits also be related to reductions in semantic fluency or narrative skills?

We have now added a detailed analysis of internal and external details for both past and future events in vmPFC patients and controls. Replicating previous findings (Bertossi et al., 2016), both patients tested in Italy and in Canada were equally impaired at producing internal details for past and future events, though in the Italian sample, both vmPFC patients and healthy controls generally performed worse for future than past events (fewer internal and external details).

P. 17 now reads: “As for the Italian sample, an ANOVA on the details produced with Group (vmPFC patients, healthy controls), Time (Past, Future), and Detail (internal, external) as factors showed a significant effect of Time (F_1,13_ = 14.66, p = 0.002, partial η^2^ = 0.53), such that all participants produced more details for past than future events (18.19 vs. 15.37). There were also significant effects of Group (F_1,13_ = 6.16, p = 0.02, partial η^2^ = 0.32) and Detail (F_1,13_ = 9.14, p = 0.009, partial η^2^ = 0.41), qualified by a Group x Detail interaction (F_1,13_ = 8.99, p = 0.01, partial η^2^ = 0.40). Post hoc Fisher tests showed that vmPFC patients produced fewer internal details (11.45 vs. 25.51; p = 0.004) but a similar number of external details than controls (11.39 vs. 11.96; p = 0.89). No other effect was significant (p > 0.31 in all cases). The same ANOVA on the Canadian sample revealed an effect of Group (F_1,22_ = 17.76, p = 0.0003, partial η^2^ = 0.44), qualified by a significant Group x Detail interaction (F_1,22_ = 4.72, p = 0.04, partial η^2^ = 0.18), again indicating that vmPFC patients produced fewer internal details (10.63 vs. 31.78; p = 0.0003) but a similar number of external details than controls (16.79 vs. 25.65; p = 0.09). No other effect was significant (p > 0.32 in all cases).”

We find it unlikely that vmPFC patients’ impairment in episodic remembering and future thinking is due to problems in fluency or narrative skills, because the deficit was limited to internal details, while patients were otherwise capable of producing external details. Moreover, all patients but one had preserved verbal fluency (Table 1). Please also note that in a previous study investigating EFT in vmPFC patients, we found that narrative deficits did not account for EFT deficits in vmPFC patients (Bertossi et al., 2017).

[Editors' note: further revisions were suggested prior to acceptance, as described below.]

The manuscript has been improved but there is one remaining issue that needs to be addressed, as outlined below:Responses to EFT cues were self-paced and subjects were instructed to respond within 10-15 seconds. It seems important to know whether there were any group differences in the time spent with the EFT cues. For instance, if patients took more time, this could indicate an actual impairment in EFT that failed to affect choices because it was compensated by taking more time. Please include an analysis of the responses time data in the manuscript, and, in case there are significant differences, discuss these effects.

We did not collect the time spent on the EFT cue screen, so we cannot add the analysis you requested. We asked participants to do their best and imagine the future event in detail for 15 seconds. Participants, including patients, largely complied with these instructions. However, we preferred to have the EFT screen self-paced and allow slight variability in imagination times (10-15 seconds, see p. 19) to make sure participants passed on to the decision screen when they still had the event active and (relatively) detail-rich in mind. We have learned from previous and pilot studies that having individuals think about a future event for a fixed amount of time carries the risk that at some point they stop thinking about the event, or revert to think about a gist (semantic) representation of the event. This is especially the case for vmPFC patients, who have difficulties holding events in working memory (Bertossi et al., 2017). Indeed, in this task, vmPFC patients, if anything, tended to prefer spending less – not more – time imagining the future, occasionally noting that “they were done imagining”. Yet, EFT reduced DD also in the vmPFC group. Also, some events may be shorter in duration than others and, therefore, enforcing fixed imagination times may in fact lead to changes in the type of processes/representations that take place during the imagination period, and immediately before the intertemporal choice. Allowing participants to pass on to the decision screen even slightly before the recommended 15 seconds maximized the possibility that they approached the intertemporal choice while still having a relatively context-rich representation of the relevant future event in mind.

We agree that having the exact imagination times would be informative, but, in general, we have no control over the time participants actually spend imagining the future event during the EFT time period and, even if we did, or we were able to measure response times, we would still not be able to gauge the type of future representations that participants are actually contemplating during that time (e.g., context-rich vs. gist like). As anticipated, this holds even in the most controlled situation one can think of – namely, with fixed imagination times.

We have added a discussion of this point on p. 14, which now reads: “Finally, our interpretation of vmPFC patients’ preserved EFT effect as due to the external cueing of semantic structures driving EFT is speculative at this point. Indeed, this study does not provide direct insight into the type of future representations underlying the EFT effect on DD in vmPFC patients and in healthy controls. However, it is consistent with evidence that vmPFC patients produce few internal (episodic) details but a normal number of external (semantic) details during EFT tasks. It is also consistent with current models of vmPFC that postulate this structure is involved in the self-initiation of event construction (e.g., McCormick et al., 2018; Ciaramelli et al., 2019; Verfaellie et al., 2019). Further work should study EFT performance in vmPFC patients under conditions that (externally) promote the selection of self-relevant cues for EFT (as in the present study) vs. those that do not, and investigate the quality and quantity of future event details that are necessary and sufficient to influence DD in vmPFC patients and in healthy controls.”